# Reproducible Bioinformatics Analysis Workflows for Detecting *IGH* Gene Fusions in B-Cell Acute Lymphoblastic Leukaemia Patients

**DOI:** 10.3390/cancers15194731

**Published:** 2023-09-26

**Authors:** Ashlee J. Thomson, Jacqueline A. Rehn, Susan L. Heatley, Laura N. Eadie, Elyse C. Page, Caitlin Schutz, Barbara J. McClure, Rosemary Sutton, Luciano Dalla-Pozza, Andrew S. Moore, Matthew Greenwood, Rishi S. Kotecha, Chun Y. Fong, Agnes S. M. Yong, David T. Yeung, James Breen, Deborah L. White

**Affiliations:** 1Faculty of Health and Medical Sciences, University of Adelaide, Adelaide, SA 5005, Australia; jacqueline.rehn@sahmri.com (J.A.R.); sue.heatley@sahmri.com (S.L.H.); laura.eadie@sahmri.com (L.N.E.); elyse.page@sahmri.com (E.C.P.); barb.mcclure@unisa.edu.au (B.J.M.); agnes.yong@health.wa.gov.au (A.S.M.Y.); david.yeung@adelaide.edu.au (D.T.Y.); deborah.white@sahmri.com (D.L.W.); 2Blood Cancer Program, Precision Cancer Medicine Theme, South Australian Health & Medical Research Institute (SAHMRI), Adelaide, SA 5000, Australia; caitlin.schutz@sahmri.com; 3Australian and New Zealand Children’s Oncology Group (ANZCHOG), Clayton, VIC 3168, Australia; 4Molecular Diagnostics, Children’s Cancer Institute, Kensington, NSW 2750, Australia; rsutton@ccia.org.au; 5The Cancer Centre for Children, The Children’s Hospital at Westmead, Westmead, NSW 2145, Australia; luciano.dallapozza@health.nsw.gov.au; 6Oncology Service, Children’s Health Queensland Hospital and Health Service, Brisbane, QLD 4101, Australia; andy.moore@health.qld.gov.au; 7Child Health Research Centre, The University of Queensland, Brisbane, QLD 4000, Australia; 8Department of Haematology and Transfusion Services, Royal North Shore Hospital, Sydney, NSW 2065, Australia; matthew.greenwood@health.nsw.gov.au; 9Faculty of Health and Medicine, University of Sydney, Sydney, NSW 2006, Australia; 10Department of Clinical Haematology, Oncology, Blood and Marrow Transplantation, Perth Children’s Hospital, Perth, WA 6009, Australia; rishi.kotecha@health.wa.gov.au; 11Leukaemia Translational Research Laboratory, Telethon Kids Cancer Centre, Telethon Kids Institute, University of Western Australia, Perth, WA 6009, Australia; 12Curtin Medical School, Curtin University, Perth, WA 6845, Australia; 13Department of Clinical Haematology, Austin Health, Heidelberg, VIC 3083, Australia; chun.fong@austin.org.au; 14South Australian Health & Medical Research Institute (SAHMRI), Adelaide, SA 5000, Australia; 15Division of Pathology & Laboratory, University of Western Australia Medical School, Perth, WA 6009, Australia; 16Department of Haematology, Royal Perth Hospital, Perth, WA 6000, Australia; 17Haematology Department, Royal Adelaide Hospital and SA Pathology, Adelaide, SA 5000, Australia; 18Black Ochre Data Labs, Indigenous Genomics, Telethon Kids Institute, Adelaide, SA 5000, Australia; 19James Curtin School of Medical Research, Australian National University, Canberra, ACT 2601, Australia; 20Australian Genomics Health Alliance (AGHA), The Murdoch Children’s Research Institute, Parkville, VIC 3052, Australia

**Keywords:** B-cell acute lymphoblastic leukaemia, *IGH*, gene fusion, workflow

## Abstract

**Simple Summary:**

B-cell acute lymphoblastic leukaemia (B-ALL) is a haematological malignancy driven by diverse genomic alterations, the most common being gene fusions, which can be detected via transcriptomic analysis. However, detecting gene fusions involving the Immunoglobulin Heavy Chain (*IGH*) region can be challenging due to its hyper variability. Our aim was to develop a workflow containing the algorithms FusionCatcher, Arriba, and STAR-Fusion, to achieve best-case sensitivity for *IGH* gene fusion detection. We analysed 35 patient samples harbouring *IGH* gene fusions and assessed the detection rates for each caller, before optimising the parameters to enhance sensitivity for *IGH* fusions. FusionCatcher and Arriba outperformed STAR-Fusion; however, by adjusting specific filtering parameters, we were able to improve STAR-Fusion’s performance. This analysis highlights the importance of filtering optimization for *IGH* gene fusion events, offering alternative workflows for difficult-to-detect high-risk B-ALL alterations.

**Abstract:**

B-cell acute lymphoblastic leukaemia (B-ALL) is characterised by diverse genomic alterations, the most frequent being gene fusions detected via transcriptomic analysis (mRNA-seq). Due to its hypervariable nature, gene fusions involving the Immunoglobulin Heavy Chain (*IGH*) locus can be difficult to detect with standard gene fusion calling algorithms and significant computational resources and analysis times are required. We aimed to optimize a gene fusion calling workflow to achieve best-case sensitivity for *IGH* gene fusion detection. Using Nextflow, we developed a simplified workflow containing the algorithms FusionCatcher, Arriba, and STAR-Fusion. We analysed samples from 35 patients harbouring *IGH* fusions (*IGH::CRLF2* n = 17, *IGH::DUX4* n = 15, *IGH::EPOR* n = 3) and assessed the detection rates for each caller, before optimizing the parameters to enhance sensitivity for *IGH* fusions. Initial results showed that FusionCatcher and Arriba outperformed STAR-Fusion (85–89% vs. 29% of *IGH* fusions reported). We found that extensive filtering in STAR-Fusion hindered *IGH* reporting. By adjusting specific filtering steps (e.g., read support, fusion fragments per million total reads), we achieved a 94% reporting rate for *IGH* fusions with STAR-Fusion. This analysis highlights the importance of filtering optimization for *IGH* gene fusion events, offering alternative workflows for difficult-to-detect high-risk B-ALL subtypes.

## 1. Introduction

B-cell acute lymphoblastic leukaemia (B-ALL) is a haematological malignancy that results in the over-proliferation of lymphoblasts [1]. While it is primarily diagnosed in children <5 years, younger patients tend to have a better prognosis than adolescents and adults diagnosed with the disease [2,3,4]. A number of genomic alterations have been identified as drivers of this disease, the most frequent being gene fusions secondary to chromosomal rearrangements [5,6], making them potential targets for therapeutic intervention and personalised medicine [6]. The accurate and rapid identification of these genomic alterations is therefore a crucial aspect of B-ALL diagnosis and treatment [7].

Prior to the introduction of Next-Generation Sequencing (NGS) and transcriptomic analysis, the detection of prognostically significant gene fusions with well-characterised breakpoints was performed using diagnostic analysis, such as fluorescence in situ hybridisation (FISH), reverse transcriptase polymerase chain reaction (RT-PCR) and G-banded karyotyping [7,8,9]; however, these processes are time-intensive, expensive, and inadequate for detecting novel and cytogenetically cryptic rearrangements [10,11]. With the implementation of transcriptome sequencing (mRNA-seq) and advanced bioinformatic analysis techniques, it is now possible to identify most fusion transcripts, including cytogenetically cryptic transcripts known to confer prognostic significance in a more time- and cost-effective manner [12,13].

Gene fusion detection via mRNA-seq involves identifying instances in which sequences from two distinct genes or genomic regions are combined, typically through structural variations, such as chromosomal rearrangements or translocations (Figure 1A). When sequencing reads are mapped to the reference genome, gene fusions are indicated by the presence of spanning and chimeric junction reads, which extend across the breakpoints in which the fusion occurs [8] (Figure 1B). To identify these transcripts, fusion calling algorithms analyse the sequencing data to detect and characterise events. Once potential fusion transcripts are identified, custom filters are applied to minimize the number of false positives caused by read alignment errors or technical artifacts. The filtering criteria implemented by fusion calling algorithms play a crucial role in reducing false positives. These criteria are often tailored to the specific characteristics of fusion events, such as read coverage, breakpoint sequence patterns, and the presence of discordant mate pairs. Some of these strict filtering guidelines, however, can result in important driver mutations not being reported. Additionally, evaluating highly repetitive and hypervariable regions of the genome, such as the immunoglobulin heavy (*IGH*) chain, remains a challenge for these algorithms [14,15].

While relying on spanning reads and junction reads is adequate for confirming most gene fusion events that occur at exon–exon boundaries, this is not the case for *IGH* fusions. Located in the sub-telomeric region of chromosome 14q, *IGH* is responsible for producing an individual’s highly variable repertoire of antibodies and comprises multiple gene regions [16]. These gene regions undergo somatic recombination during early B-cell differentiation, resulting in an *IGH* gene rearrangement [17,18]. The aberrant recombination of *IGH* can result in a gene or gene segment from another chromosomal location being spliced upstream of an *IGH* enhancer, resulting in the upregulation of the partner gene, which acts as an oncogenic driver. A consequence of this aberrant recombination is the insertion of non-template nucleotides at fusion breakpoints [17]. These additional non-template nucleotides, along with the limitations of short-read sequencing technologies [15,16,19,20], often result in suboptimal read mapping across the *IGH* fusion breakpoints, leading to the under-reporting of these fusions by standard diagnostic and transcriptomic analysis methods.

*IGH* fusions also have a diverse range of partner genes [21,22,23]. Whilst the biological significance and clinical implications vary depending on the partner gene, many *IGH* fusions are indicative of a poor prognosis. In 2014, Russel et al. [4] successfully identified 14 different *IGH* partner genes, including *CRLF2* and *EPOR,* across 75 B-ALL patients, all of which were associated with high-risk subtypes. However, the partner gene remained unknown for a further 84 patients who tested positive for an *IGH* translocation via FISH probe. As a result of the increased detection sensitivity through the implementation of mRNA-seq analysis, the number of successfully identified *IGH* partner genes has risen in the intervening years to also include the cytogenetically cryptic *DUX4* [2,5,24].

The identification of gene fusions in patient samples in a time-efficient manner is crucial for effective risk stratification and personalised treatment planning. Current mRNA-seq analyses are time-consuming, at times exceeding 20 h to analyse one sample [25], and require significant personnel, specialised expertise, and computational resources to establish and execute them in a routine clinical context. In recent years, the Bioinformatics and Clinical Genomics community has attempted to address these issues by using state-of-the-art analysis workflow tools that can be implemented in routine sequencing and research labs [26,27]. However the careful consideration of parameter and filtering selection may still be required to achieve detection sensitivity to potential gene fusion events. In this study, we developed a reproducible mRNA-seq analysis workflow to accurately identify B-ALL gene fusions using the Nextflow workflow system [28]. We assessed the pipeline’s accuracy, speed, and computational requirements when analysing the B-ALL patient samples harbouring *IGH* fusions. By assessing these difficult-to-detect fusions through mRNA-seq analyses, we aim to identify potential areas for improvement in sensitivity and accuracy.

## 2. Materials and Methods

The *RIGHT* (Recovering *IGH* fusion Transcripts) workflow (Figure 2) was developed using the workflow management executor Nextflow, version 21.10.6 [28]. Nextflow facilitates the parallel resolution of the implemented algorithms, namely STAR-Fusion, version 1.10.0 [24], Arriba, version 2.1.0 [25], and FusionCatcher, version 1.33 [29], thus optimising the processing time and enabling reproducibility across distributed computing infrastructures. STAR-Fusion and Arriba are often used together owing to their high accuracy and short execution times on both simulated and real data [24,25]. Adherence to the recommended best practice for mRNA-seq data analysis involves employing at least two distinct gene fusion calling algorithms, with intersected results for improved accuracy [25]. The GRCh37 human reference genome [30,31] was used with the Ensemble reference transcript annotation [32].

To evaluate the ability of the workflow to efficiently analyse *IGH* fusions, a subset of 35 Australian B-ALL patient samples that were referred to the South Australian Health and Medical Research Institute (SAHMRI) (Adelaide, Australia) for genomic testing were selected (Appendix A), all of which were confirmed to harbour *IGH* fusions (*IGH::CRLF2* n = 17, *IGH::DUX4* n = 15, *IGH::EPOR* n = 3). These 3 gene fusions were chosen as they are the most common *IGH* fusions seen at SAHMRI. *IGH::CRLF2* fusions were confirmed via fluorescent in situ hybridization (FISH) using two separate break-apart probes, a Vysis IGH dual colour probe (Abbott) and a *CRLF2* dual colour probe (Cytocell). Gene expression profiling was used to confirm *IGH::DUX4* fusions, and the visual inspection of BAM files in the Integrative Genomics Viewer (IGV) was needed to confirm the presence of *IGH::EPOR* (Appendix A). We also analysed a second subset of 25 B-ALL patient samples that contained non-*IGH* gene fusion events, and non-translocated samples (Appendix A) as a control. These 60 samples were chosen from a previously published cohort of 180 B-ALL patient samples [33] that represented a range of common genomic lesions associated with ALL subtypes (EGA Accession: EGAS00001006460) [34]. The process of preparing libraries for mRNA sequencing was carried out using either the TruSeq Stranded mRNA LT Kit (Illumina, CA, USA) or the Universal Plus mRNA-Seq with NuQuant (Tecan, CA, USA). This was performed using 400 ng of total RNA, as per the instructions provided by the manufacturers. Subsequently, the samples underwent sequencing on either the Illumina HiSeq 2000 or NextSeq 500 platforms. The outcome of this sequencing approach yielded paired-end (PE) reads with a length of 75 bases, and an average read depth of 70 million reads.

To assess the sensitivity of the developed workflow, we processed the patient samples using the default parameters for each algorithm (Appendix A). Given the initial output files, we compared the reporting rates for *IGH* fusions for each caller and determined that STAR-Fusion had the lowest reporting rate. We then assessed the output files that STAR-Fusion produced following each stage of filtering to determine why the number of reported *IGH* fusions was lower than the other callers. STAR-Fusion produces intermediate output files during filtering comprising pre- and post-blast-filter files, listing each gene fusion detected and whether they passed filtering or not. If they did not pass filtering, the filtering step at which failure occurred was listed, and why. Given this, we were able to trace each *IGH* fusion and determine which filtering steps failed. We were then able to adjust the appropriate parameters in STAR-Fusion and process the samples again to reassess the reporting rates (Appendix A). While we used the GRCh37 reference genome, as this is currently the primary reference used at SAHMRI when analysing B-ALL patient samples, this pipeline is also compatible with the newer reference genome, GRCh38.

## 3. Results

### 3.1. Using RIGHT to Detect IGH Gene Fusion Events

Initially, we processed the 35 patient samples through the *RIGHT* workflow using the default parameters for each algorithm (Appendix A). Between the three algorithms, all *IGH* fusions were detected (Figure 3 and Appendix A), with 74% being detected by at least two of the tools. Arriba and FusionCatcher significantly outperformed STAR-Fusion, with the two tools successfully reporting all *IGH::EPOR* (n = 3) fusions, while the cytogenetically cryptic *IGH::DUX4* fusions (n = 15), which frequently go undetected, were reported in 93% (n = 14) of samples by Arriba and in 80% (n = 12) by FusionCatcher. In contrast, STAR-Fusion failed to detect any of the *IGH::EPOR* fusions and only reported 33% (n = 5) of the *IGH::DUX4* fusions. Of the 17 samples harbouring *IGH::CRLF2* fusions, FusionCatcher and Arriba reported 94% (n = 16) and 70% (n = 12), respectively, while STAR-Fusion again underperformed with only 29% (n = 5) of the fusions being detected.

A trade-off exists between high sensitivity and an increased number of overall predicted fusions, many of which are not clinically relevant or potential false positives. While STAR-Fusion demonstrated lower sensitivity, this was balanced with fewer total predictions, an average of 12 fusions per sample, in contrast to FusionCatcher which reported an average of 259 fusions per sample (Appendix A).

### 3.2. Improving STAR-Fusions Performance when Detecting IGH Gene Fusions

An examination of the intermediary files generated by STAR-Fusion revealed that 33 of 35 *IGH* fusions were initially predicted, but most were subsequently excluded as they did not meet one or more of the default filtering parameters. This included insufficient junction read support and/or novel junction read support (nine samples for each criterion). For example, one sample had a read support depth of 1298 spanning fragments, but no junction reads, leading to exclusion from further filtering (Appendix A). To evaluate the effect the default values for the minimum number of supporting reads had on the overall fusion reporting rate, we decreased the values of these parameters and re-ran STAR-Fusion (Figure 4, Appendix A).

The number of reported *IGH::DUX4* fusions increased following the initial parameter changes, with an almost three-fold increase (n = 14) in reported fusions when both the junction read and novel junction support read values were altered. The number of *IGH::CRLF2* fusions reported increased as well (n = 6), while *IGH::EPOR* fusions remained undetected. The overall reporting rate increased from 29% using default values to 60%, still lower than that of Arriba and FusionCatcher. Additionally, despite increasing sensitivity, the total number of fusion events detected decreased compared to the default settings, averaging 10 fusions per sample (Appendix A). It is unclear why this occurred. We attempted to trace the select fusion events reported when using default parameters that were subsequently dropped when using altered parameters, but no pattern emerged.

STAR-Fusion implements an expression filter to normalise the number of RNA-seq reads supporting a fusion breakpoint based on the overall sequencing depth, represented as fusion fragments per million total mRNA-seq fragments (FFPM). By default, a candidate fusion breakpoint must have an FFPM value greater than 0.1, equating to one supporting read per 10 million mRNA-seq reads. We found that 10 candidate fusions were subsequently rejected because of an FFPM value lower than 0.1. By reducing the minimum FFPM in combination with the lowered minimum read support, the number of *IGH::DUX4* fusions reported stayed the same (n = 14) while the *IGH::CRLF2* fusions increased (n = 12), with STAR-Fusion also reporting all *IGH::EPOR* (n = 3), for an overall reporting rate of 82.9% (n = 29) (Figure 4). We also observed a significant increase in the total number of fusion events reported by STAR-Fusion, averaging 162.4 fusions per sample (Appendix A). Despite resulting in more predicted fusions, these results were still significantly lower than those of FusionCatcher, which reported almost 9100 total fusion events across the 35 samples, averaging nearly 260 fusions per sample, while Ariba also had more predicted fusions, with an average of 48 per sample (Appendix A).

In addition to extensive parameter customization, STAR-Fusion offers pre-set parameter bundles that are designed for maximum sensitivity. To assess the impact of these pre-set bundles on *IGH* fusion reporting, we conducted two additional tests using these undocumented parameter bundles (Figure 4, Appendix A). Surprisingly, using the Max Sensitivity parameter bundle resulted in fewer *IGH::CRLF2* and *IGH::EPOR* fusions being detected than when individually reducing FFPM and read support thresholds. We also observed a considerable increase in the total number of gene fusion events reported, with a total of 44,801 reported, averaging 1280 per sample. When applying the Full Monty parameter bundle, 94% of *IGH* fusions were detected, more than that achieved with any other STAR-Fusion settings tested, but at the expense of a substantial increase in the total number of fusion events reported, which amounted to 2,464,365, an average of 70,410 per sample (Appendix A).

### 3.3. Runtime and Resources

The *RIGHT* workflow was run on a HPC cluster, comprising seven nodes, each node running Intel(R) Xeon(R) Silver 4309Y CPU (@2.80 GHz) with 32 threads and 128 GB of memory, with Ubuntu 20.04.5 LTS as its operating system. *RIGHT* offers a runtime that is solely dependent on the slowest algorithm, by allowing all tools to be executed in parallel and independently of each other. Among the algorithms used, Arriba demonstrated the shortest runtime, averaging 15 min per sample (Table 1 and Appendix A) due to its one-pass filtering technique and its ability to extract candidate reads during the alignment process. FusionCatcher displayed the lengthiest average runtime of 2 h 39 min, while the default STAR-Fusion averaged 55 min. Memory consumption was comparable between each algorithm, with FusionCatcher using the least memory, with an average of 24.9 GB, while STAR-Fusion and Arriba used slightly more memory, at 38 GB and 35.8 GB, respectively. Finally, the modifications made to enhance STAR-Fusion’s sensitivity only marginally increased its runtime and resource usage (Appendix A). At most, the average runtime increased by 8 min per sample for the Full Monty parameter bundle, while the average memory usage remained between 38.2–38.5 GB.

### 3.4. Ensemble Filtering Approaches to Identify IGH Fusions

The current recommended best practice for RNA-seq data analysis involves employing a minimum of two different gene fusion calling algorithms and reporting the intersected results [25]. We therefore compared the proportion of *IGH* fusions detected by multiple callers using both the default STAR-fusion settings and the more sensitive Full Monty setting (Figure 5).

The default *RIGHT* workflow results in 74.3% (n = 26) of *IGH* gene fusion events being reported by two or more callers, but this decreased to 28.6% (n = 10) when detected by all three callers (Figure 5A). When using STAR-Fusion with the Full Monty bundle, the number of gene fusion events detected by all algorithms increased to 71.4% (n = 25), with 97.1% (n = 34) detected by at least two callers (Figure 5B). While this is a marked improvement, the pipeline was still limited by the slowest-running algorithm, FusionCatcher. We then compared this result to only the intersection of the two fastest algorithms, in a bid to decrease the overall processing time. By using only Arriba and STAR-Fusion Full Monty, we saw 85.3% (n = 29) of gene fusion events reported by both, while also decreasing the average processing time by 1 h 36 min per sample. Given this, it can be argued that using only Arriba and the optimised STAR-Fusion, and reporting the intersection of their results, can lead to reduced processing times and improved reporting when compared to the initial default pipeline, while still adhering to the current best practice. If the processing time is not of concern, the use of all three callers is recommended; however, the combination of Arriba and FusionCatcher is also a practical option, and results in 74.3% of fusions being detected by both callers. Regardless of the combination of algorithms used, all *IGH* fusions present in the samples are detected by at least one of the callers (Appendix A).

To ensure that the STAR-Fusion parameter optimisation that was implemented to improve the detection rate of *IGH* gene fusions did not affect the detection rate of non-*IGH* gene fusion and non-translocated cases, we processed 25 samples through the default workflow and the optimised workflow. These samples represented a range of B-ALL subtype defining alterations, including *ETV6::RUNX1*, *BCR::ABL1*, *PAX5::JAK2, KMT2A::AFF1* and *EP300::ZNF384* gene fusions, as well as PAX5.P80R and IKZF1.N159Y (Appendix A). For these 25 samples, all three algorithms with default parameters, as well as the optimised STAR-Fusion, found all 19 expected gene fusion events (Appendix A), and no pathogenic fusions were reported for samples that did not harbour translocation events. To ensure no additional pathogenic gene fusion events were reported when using the optimised STAR-Fusion parameters, we filtered the results against an extensive list of known pathogenic translocation events seen in B-ALL (Appendix A), and no additional pathogenic gene fusions were reported.

## 4. Discussion

The accurate and rapid identification of gene fusion events is crucial for assessing a patient’s prognosis to facilitate efficient risk stratification and treatment. As there is no single universally accepted gold standard tool for gene fusion detection, current transcriptomic analysis approaches rely on the use of multiple algorithms and tools [8], resulting in a time-consuming process that requires significant personnel and computational resources to establish and execute in a routine clinical context. With this, we have provided a best-case computational pipeline for detecting gene fusion events in B-ALL patient samples, with a particular focus on those involving the hypervariable and highly repetitive *IGH* locus, which are notoriously difficult to detect, using a combination of specific algorithms and parameters.

The default parameters employed by standard gene fusion detection algorithms generally demonstrate adequacy in identifying gene fusion events in B-ALL, as is shown here when analysing patient samples that harbour translocation events involving distinct exon-exon breakpoints (Appendix A). However, our research here has also shown that *IGH* gene fusion events, owing to their highly variable nature, necessitate a more refined and sensitive parameter set. Our findings demonstrate that by adjusting merely three parameters, STAR-Fusion achieved a substantial enhancement in *IGH* fusion detection (94%), surpassing both Arriba and FusionCatcher (85.7% and 88.6%, respectively). This remarkable improvement can be attributed to the insufficiency of standard minimum filtering parameters, which fail to exhibit the necessary sensitivity for detecting many of these events due to the absence of distinct exon-exon breakpoints in *IGH* fusions. However, the trade-off between sensitivity and specificity must be considered, as higher sensitivity can also lead to an increase in prospective fusions, which would be unfeasible to manually curate. Moreover, despite the potential value of identifying all gene fusion events in a given patient sample, the number of clinically actionable fusions remains low [35], making it crucial to strike a balance between filtering sensitivity and accuracy, so that prognostically significant fusions are not missed and false positives are limited.

Reducing the time taken for disease classification is imperative when undertaking risk assessment and personalised patient treatment planning. Ensemble detection approaches using Nextflow allow tools to be run in parallel, reducing the overall processing time. As seen here, Arriba had an average runtime of 15 min, STAR-Fusion’s runtime averaged 55 min, while FusionCatcher’s average runtime was 2 h 39 min. However, by parallelising these processes, the time is only limited by the longest running algorithm. Even the changes made to STAR-Fusion increased its average runtime by only 8 min. And yet, we showed that it was possible to use only Arriba and the optimised STAR-Fusion and obtain a higher reporting rate of *IGH* gene fusion events detected by at least two algorithms compared to the default pipeline (85.3% vs. 74.3%) (Figure 5), while decreasing the processing time to 1 h per sample. While this time difference may seem negligible for an individual sample, processing multiple samples simultaneously results in a substantial improvement.

The identification of gene fusion events, particularly high-risk *IGH* fusions, is critical for B-ALL treatment and research, as they are often therapeutically targetable and frequently associated with the development and progression of various subtypes of the disease. Therefore, selecting the best-case parameters for a given algorithm, in order to achieve the most clinically significant results, is of paramount importance. However, while manually testing combinations of parameters, as seen here, can be time-consuming, machine learning and deep learning algorithms are rapidly evolving and have the potential to provide a more efficient means of identifying biomarkers in multi-omics datasets for personalised medicine [36,37]. Ultimately, future directions for precision medicine in B-ALL may involve the use of machine learning to identify the specific diagnostic markers present in an individual patient sample, and automatically select the optimal parameters for a given diagnostic algorithm.

In the present study, while our findings show promising insights into the detection of *IGH* gene fusions through bioinformatic analysis, it is imperative to address some limitations that have potential implications for the interpretation of our results. The use of short-read sequencing technology presents an acknowledged constraint. Short reads have been shown to exhibit lower accuracy in comparison to long reads [38,39], potentially impacting the sensitivity and specificity of *IGH* gene fusion detection. Additionally, the rapidly evolving landscape of bioinformatics introduces the consideration of newer gene fusion calling algorithms that might offer enhanced accuracy and sensitivity. The evaluation and incorporation of such algorithms, specifically those that have been tailored for the detection of difficult-to-identify gene fusions, warrant exploration to comprehensively assess their performance in our context [40,41,42].

## 5. Conclusions

The accurate and rapid detection of gene fusions is vital for assessing prognosis and guiding treatment in B-ALL. However, the lack of a universally accepted detection tool necessitates using multiple algorithms, leading to a time-consuming process in clinical practice. Our study showed that through curated parameter values for specific algorithms, detecting gene fusions involving the hypervariable *IGH* locus can be improved when compared to default values. However, balancing sensitivity and specificity is crucial to avoid a disproportionate number of fusions that are not clinically relevant. By using an ensemble detection approach and Nextflow, the workflow manager, we increased the detection rate of *IGH* fusions and significantly reduced the processing time. With the use of optimised algorithms determined through machine learning, the future holds promise for the efficient identification of biomarkers and parameter selection in personalised medicine for B-ALL.

## Figures and Tables

**Figure 1 cancers-15-04731-f001:**
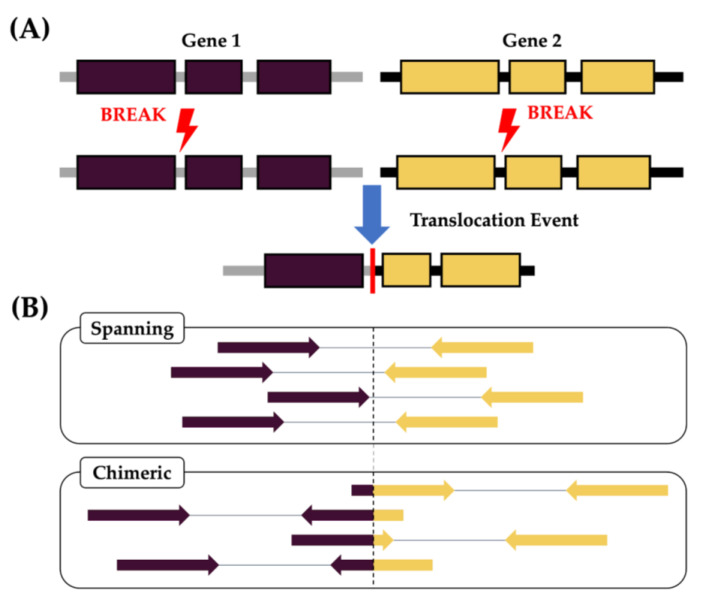
(**A**) Schematic of a gene fusion event, in which two separate genes located on the same or different chromosomes become juxtaposed, following a chromosomal rearrangement or translocation event. (**B**) Fusion events can be identified through the alignment of paired-end mRNA-seq reads to the reference genome, when each end of the paired read maps entirely to two different genomic regions (Spanning) or when one end of the read overlaps the fusion breakpoint (Chimeric).

**Figure 2 cancers-15-04731-f002:**
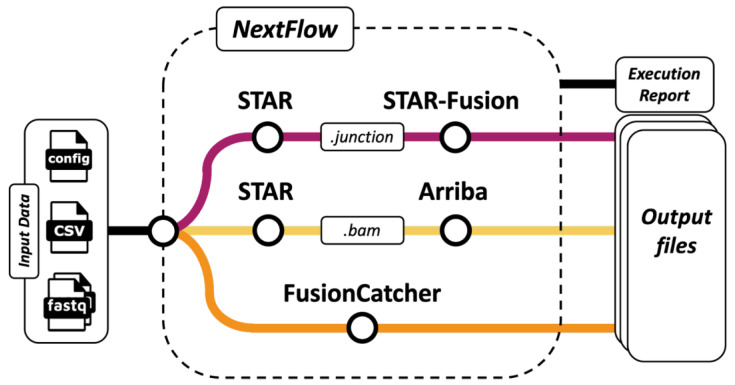
Schematic of the *RIGHT* workflow. STAR-Fusion, Arriba and FusionCatcher algorithms are all executed using the Nextflow system, taking an initial. csv sample manifest as input, which contains directory links and file base names for paired-end RNA-seq FASTQ files. The config file contains information regarding the resources requested, the scheduler used, e.g., Conda, Slurm, and any container options, such as Docker or Singularity. Arriba and STAR-Fusion have different formatting requirements for their alignment inputs, necessitating two separate STAR alignment runs.

**Figure 3 cancers-15-04731-f003:**
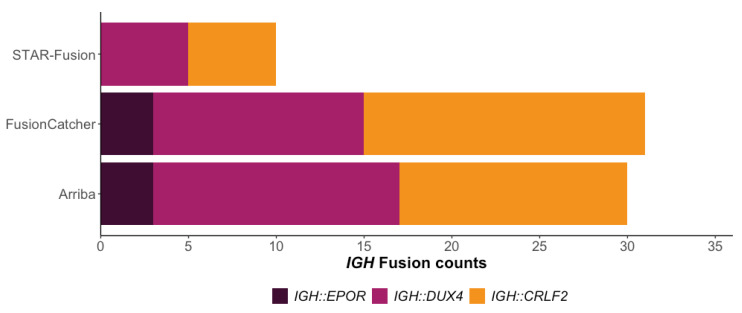
The number of *IGH* fusion events reported by each algorithm using default parameters (*IGH::CRLF2* n = 17, *IGH::DUX4* n = 15, *IGH::EPOR* n = 3).

**Figure 4 cancers-15-04731-f004:**
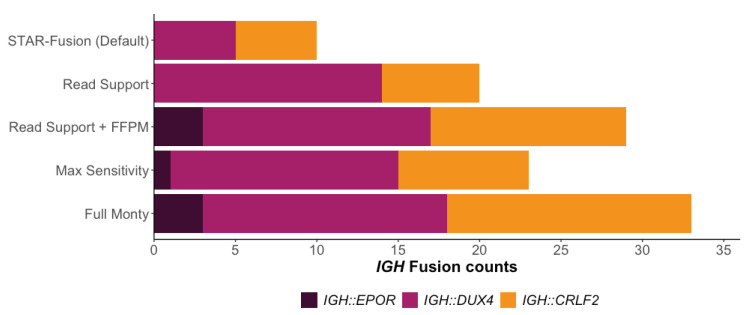
The number of *IGH* gene fusions reported by STAR-Fusion with the default parameter values vs. the altered parameter values. Read Support includes novel junction support reads and junction support reads, and FFPM is the fusion fragments per million reads.

**Figure 5 cancers-15-04731-f005:**
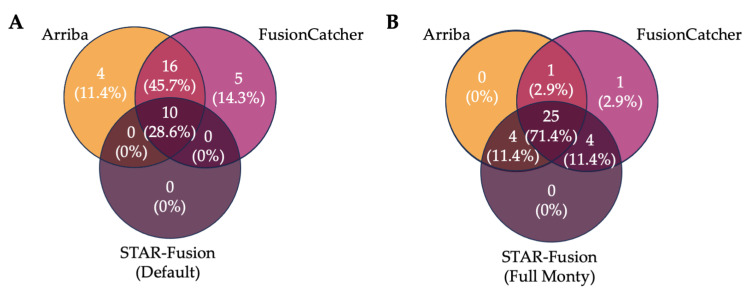
The intersection of gene fusion events reported by each algorithm, from (**A**) the full default pipeline, (**B**) Arriba, FusionCatcher, and the best-performing STAR-Fusion test (Full Monty).

**Table 1 cancers-15-04731-t001:** The maximum, minimum and mean values for the time taken for each algorithm to run, and the resident set size (RSS) required.

	Time			RSS (GB)		
	Max	Mean	Min	Max	Mean	Min
Arriba	27 m 44 s	15 m	6 m 43 s	38.2	35.8	34.5
FusionCatcher	4 h 25 m 21 s	2 h 39 m	1 h 23 m 50 s	27.5	24.9	22.5
STAR-Fusion (Default)	1 h 38 m 6 s	55 m	27 m 37 s	39.2	38.3	37.5
Read Support	1 h 33 m 6 s	57 m	26 m 35 s	39.2	38.3	37.4
Read Support + FFPM	1 h 29 m 18 s	56 m	27 m 4 s	39.2	38.3	37.4
Max Sensitivity	2 h 11 m 36 s	1 h 3 m	26 m 21 s	39.2	38.3	37.5
Full Monty	2 h 11 s	1 h 3 m	26 m 52 s	39.2	38.5	37.4

## Data Availability

Genome–phenome Archive (accession numbers EGAD00001004461 and Raw FASTQ files for ALL samples in the study cohort are available from Genome-phenome Archive (EGAS00001006460). Code and additional scripts needed to reproduce the method seen here can be found in the Git repository (https://github.com/ashleethomson/The-RIGHT-Workflow).

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
