# Peer review of "Reproducible Bioinformatics Analysis Workflows for Detecting IGH Gene Fusions in B-Cell Acute Lymphoblastic Leukaemia Patients"

_cancers, 2023, doi:10.3390/cancers15194731_

Round 1

Reviewer 1 Report (Previous Reviewer 2)

Authors have adequately answered to the raised issues. No further comments.

Reviewer 2 Report (Previous Reviewer 3)

All issues have been properly addressed.

This manuscript is a resubmission of an earlier submission. The following is a list of the peer review reports and author responses from that submission.

Round 1

Reviewer 1 Report

This manuscript describes the analysis of 3 different fusion gene detection software tools. Using a truth set of 34 B-ALL patients with known IGH fusion genes and tuning the filters of different tools to accurately detect, while balancing specificity and sensitivity. They compare the computational resources utilized by each tool. They have a NextFlow pipeline they have made available. The paper is well written but has a few limitations such as only using one dataset to tune the parameters.

Major:

1)      Only one dataset is used to tune the parameters of the tools. It would be good to test these parameters on another dataset to make sure that these parameters are not tuned just for this dataset.

2)      Will it work with newer reference genomes such as GRCh38?

3)      How does this manuscript differ from “Fusion InPipe, an integrative pipeline for gene fusion detection from RNA-seq data in acute pediatric leukemia”?

Minor:

1)      Why only focus on 3 IGH gene fusions?

2)      Does this affect the identification of other non-IG gene fusions?

3)      Is it only tuned for IGH or would this improve the identification of other types of IG (IGK, IGL) gene fusions?

4)      Does STAR-Fusion the only tool that allows parameter tuning?

5)      Detail the limitations of this study.

Reviewer 2 Report

In the manuscript, Thompson and colleagues compare three algorithms for calling IGH translocations in B-ALL, reporting the overlap of the results and an interesting optimization of one of them. Although being well-written, the results appear quite circumstantial and not fully convincing.

- Authors mainly focus on the expected translocation in each sample; do the different algorithms, especially the improved STAR-Fusion, detect also other translocations? Can therefore be a false positive score as well?

- The fact that some translocations are still unreported is troublesome in a diagnostic workflow; Conclusions drawn from Figure 5C would report 6 cases without IGH translocation, although being certified as IGH-translocated. In perspective,

- How the three algorithms perform in non-translocated cases? Authors correctly comment on the dangers of a disproportionate detection of false positives, which is however not investigated.

- Line 284, percentages of 91 and 75% have possibly been swapped?

- By which assay the samples were diagnosed as IGH-translocated? Also, which library prep kit was used for the original RNAseq?

- Line 291: I would suggest the wording “improved” or “optimized” instead of “altered”.

Reviewer 3 Report

line 225: Additionally, despite increasing sensitivity, the total number of fusion events detected decreased compared to the default settings, averaging 10 fusions per sample. (Supplementary Table 6)

This is surprising. Do you have any explaination for that?

line 259:

Which hardware was used? Laptop, desktop, cluster?

line 285: 71.4% (n=25) detected by at least two callers (Figure 5B)

Figure 5B shows 71.4 % are detected by three callers

percentages of Figure 5C should also have the full 35 events as base, not only the 34 that were detected with the 2 algorithms.

line 314: surpassing both Arriba and FusionCatcher (24%)

I did not find where the 24% come from.

line 319: ... which can be time-consuming to manually curate

I would rather call it unfeasable to manually curate ten thousands of events.

I don't see a benefit of using the "Full Monty" setting.

If I understand it correctly, it is only used to confirm the known fusion events already identified by two other callers, at the cost of introducing a massive amount of false positives to the analysis.

Why not just use Arriba and FusionCatcher, which hava a 74% intersection in detected events?

Please provide reasoning for that.

Overall, the article is very well written. It reads fluently and the graphics are easy to understand and aesthetically pleasing.

Computational analysis has become an integral component of modern scientific investigations.

Yet, there are very few "gold standard" tools and the scientific community greatly benefits from publications where the capabilities and drawbacks of bioinformatic tools are investigated.